# Anticoagulants as Potential SARS-CoV-2 M^pro^ Inhibitors for COVID-19 Patients: In Vitro, Molecular Docking, Molecular Dynamics, DFT, and SAR Studies

**DOI:** 10.3390/ijms232012235

**Published:** 2022-10-13

**Authors:** Ayman Abo Elmaaty, Wagdy M. Eldehna, Muhammad Khattab, Omnia Kutkat, Radwan Alnajjar, Ahmed N. El-Taweel, Sara T. Al-Rashood, Mohammed A. S. Abourehab, Faizah A. Binjubair, Mohamed A. Saleh, Amany Belal, Ahmed A. Al-Karmalawy

**Affiliations:** 1Department of Medicinal Chemistry, Faculty of Pharmacy, Port Said University, Port Said 42526, Egypt; 2Department of Pharmaceutical Chemistry, Faculty of Pharmacy, Kafrelsheikh University, Kafrelsheikh 33516, Egypt; 3School of Biotechnology, Badr University in Cairo, Badr City 11829, Egypt; 4Department of Chemistry of Natural and Microbial Products, Pharmaceutical and Drug Industries Research Institute, National Research Centre, El-Buhouth St., Dokki, Cairo 12622, Egypt; 5Center of Scientific Excellence for Influenza Viruses, National Research Centre, Giza 12622, Egypt; 6Department of Chemistry, Faculty of Science, University of Benghazi, Benghazi 16063, Libya; 7PharmD, Faculty of Pharmacy, Libyan International Medical University, Benghazi 16063, Libya; 8Department of Chemistry, University of Cape Town, Rondebosch 7701, South Africa; 9Department of Pharmaceutical Chemistry, College of Pharmacy, King Saud University, Riyadh 11451, United Arab Emirates; 10Department of Pharmaceutics, Faculty of Pharmacy, Umm Al-Qura University, Makkah 21955, United Arab Emirates; 11Department of Clinical Sciences, College of Medicine, University of Sharjah, Sharjah 27272, United Arab Emirates; 12Department of Pharmacology and Toxicology, Faculty of Pharmacy, Mansoura University, Mansoura 35516, Egypt; 13Medicinal Chemistry Department, Faculty of Pharmacy, Beni-Suef University, Beni-Suef 62514, Egypt or; 14Department of Pharmaceutical Chemistry, College of Pharmacy, Taif University, P.O. Box 11099, Taif 21944, Saudi Arabia; 15Pharmaceutical Chemistry Department, Faculty of Pharmacy, Ahram Canadian University, 6th of October City, Giza 12566, Egypt

**Keywords:** SARS-CoV-2, M^pro^, anticoagulants, in vitro, in silico

## Abstract

In this article, 34 anticoagulant drugs were screened in silico against the main protease (M^pro^) of SARS-CoV-2 using molecular docking tools. Idraparinux, fondaparinux, eptifibatide, heparin, and ticagrelor demonstrated the highest binding affinities towards SARS-CoV-2 M^pro^. A molecular dynamics study at 200 ns was also carried out for the most promising anticoagulants to provide insights into the dynamic and thermodynamic properties of promising compounds. Moreover, a quantum mechanical study was also conducted which helped us to attest to some of the molecular docking and dynamics findings. A biological evaluation (in vitro) of the most promising compounds was also performed by carrying out the MTT cytotoxicity assay and the crystal violet assay in order to assess inhibitory concentration 50 (IC_50_). It is worth noting that ticagrelor displayed the highest intrinsic potential for the inhibition of SARS-CoV-2 with an IC_50_ value of 5.60 µM and a safety index of 25.33. In addition, fondaparinux sodium and dabigatran showed promising inhibitory activities with IC_50_ values of 8.60 and 9.40 µM, respectively, and demonstrated safety indexes of 17.60 and 15.10, respectively. Moreover, the inhibitory potential of the SARS-CoV-2 M^pro^ enzyme was investigated by utilizing the SARS-CoV-2 M^pro^ assay and using tipranavir as a reference standard. Interestingly, promising SARS-CoV-2 M^pro^ inhibitory potential was attained for fondaparinux sodium with an IC_50_ value of 2.36 µM, surpassing the reference tipranavir (IC_50_ = 7.38 µM) by more than three-fold. Furthermore, highly eligible SARS-CoV-2 M^pro^ inhibitory potential was attained for dabigatran with an IC_50_ value of 10.59 µM. Finally, an SAR was discussed, counting on the findings of both in vitro and in silico approaches.

## 1. Introduction

The 2019 new coronavirus disease (COVID-19) has vastly spread around the world [1,2]. SARS-CoV-2 contributed to more than 6,287,995 deaths globally to date. More than 521,011,797 cases have been reported as confirmed infection cases [3,4,5]. Therefore, the discovery of effective anti-SARS-CoV-2 medications is vigorously needed for combating this pandemic disease [6,7,8].

Coronaviruses (CoVs) are enveloped viruses with a spherical shape carrying an array of projections on their surfaces. The main protease (M^pro^) protein was found to have a critical function in the gene expression and replication of CoV (e.g., MERS-CoV and SARS-CoV) [9,10]. The protease is involved in the conversion of large polypeptides into smaller functional protein units. M^pro^ is composed of a three-domain cysteine protease (domains I to III) incorporated in most maturation cleavage proceedings within the precursor polyprotein. Hence, the active M^pro^ is considered a homodimer with two protomers. The coronavirus M^pro^ exhibits a non-canonical Cys-His dyad which is oriented in the cleft between domains I and II. M^pro^ is conserved among coronaviruses, and substrates of M^pro^ in many coronaviruses share several similar characteristics. Hence, the cleavage site is located between P1 and P1′. Gln amino acid is a crucial requirement in the P1 position of the substrates. Therefore, the M^pro^ is an excellent antiviral target because it lacks a human homolog [11]. M^pro^ can be considered a pivotal biological target for the discovery of drugs against coronavirus. Researchers have targeted the M^pro^ protein in the literature. For example, Rangsinth et al. assessed a set of 36 natural compounds for their potential against SARS-CoV-2 main protease using in silico molecular docking and ADMET studies [12,13].

In addition, working on repurposed drugs helps to skip the preliminary drug screening processes, hence moving directly to clinical trials, which in turn can hasten FDA approval for the newly proposed indications. Drug repurposing can be a quicker and less expensive method of drug discovery than de novo drug development [14,15,16,17].

Drug repurposing is defined as the reusage of an existing drug for treating another disease that is quite different from the originally intended indication. Hence, in a short time, a new drug can become available in the market at a lower cost compared to when a de novo drug approach is used [18,19,20,21,22]. The M^pro^ is a favorable target for SARS-CoV therapeutic design, and a wide range of drug inhibitors have been designed to efficiently target it [23,24,25]. Previous analyses of genome sequence have displayed that SARS-CoV-2 engages with the corresponding SARS-CoV and MERS-CoV variants with a high level of sequence similarity [26]. Therefore, M^pro^ is regarded as a very promising biological target for SARS-CoV-2.

Notably, the induced hypercoagulability attributed to COVID-19 has been shown to play a role in COVID-19 symptoms [27]. SARS-CoV-2 triggers the release of cytokines, such as interleukins and interferons, which leads to a systemic inflammatory response and most probably a cytokine storm. A systemic thrombus generation with cerebral infarction and pulmonary artery thrombosis can be experienced at high cytokine concentrations [28]. It is worth noting that the incidence of venous thromboembolic events (VTEs) is 30% among COVID-19 patients, whereas it is 1.3% among non-COVID-19 patients [27]. A significant increase in D-dimer and prothrombin levels with a lowering in fibrinogen levels was reported in COVID-19 non-survivors on days 10–14 [29]. Elevated D-dimer values (above 1 μg/mL) are considered vigorous and distinct death risk factors [28]. Therefore, the use of an anticoagulant is found to be associated with decreased mortality in all patients, particularly those with a sepsis-induced coagulopathy score higher than 33% [29].

Scientists have been making tremendous efforts to obtain deep insights about SARS-CoV-2 as well as the pathophysiology that it may cause. The accumulation of information on SARS-CoV-2, SARS-CoV, and MERS-CoV helps scientists and researchers to recognize novel therapeutics, biological targets, and vaccines. Thus, revealing novel effective therapeutics and/or producing a vaccine is currently a time-counting race [30].

Consequently, in this paper, we aimed to target the M^pro^ protein in relation to COVID-19 treatment. Therefore, we conducted a feasible screening using a selected library of most FDA-approved anticoagulant drugs (Figure 1). The selected anticoagulants were docked using the 3D structure of a M^pro^ dimeric form (PDB ID: 6Y2G) [31]. Hence, the most promising candidates were investigated by some in vitro approaches, assuring their potential against SARS-CoV-2. Thus, we were able to investigate the most promising candidates exerting antiviral activity towards SARS-CoV-2, which can help us propose various anticoagulant drugs to be utilized in blood clotting treatment for COVID-19 patients, especially for patients with chronic heart diseases (Figure 2).

## 2. Results and Discussion

### 2.1. Molecular Docking Studies

Molecular docking simulations of the selected anticoagulants, along with the O6K inhibitor (represented in Figure 1) in the M^pro^ active pocket of SARS-CoV-2, were performed. They were stabilized into the M^pro^ binding pocket using various interactions. Their order of binding strength based on their docking score was as follows: idraparinux (**30**) > fondaparinux (**29**) > eptifibatide (**25**) > heparin (**23**) > O6K inhibitor (**35**, docked) > ticagrelor (**26**) > dipyridamole (**14**) > argatroban (**31**) > edoxaban (**21**) > tirofiban (**24**) > ximelagatran (**32**) > apixaban (**19**) > dabigatran (**20**) > cilostazol (**33**) > betrixaban (**28**) > ethyl eicosapentaenoic acid (**18**) > prasugrel (**27**) > rivaroxaban (**22**) > sulfinpyrazone (**17**) > defibrotide (**34**) > diphenadione (**11**) > ethyl biscoumacetate (**7**) > acenocoumarol (**4**) > cyclocoumarol (**6**) > dicoumarol (**2**) > warfarin (**5**) > clopidogrel (**16**) > ticlopidine (**15**) > phenprocoumon (**3**) > anisindione (**9**) > bromindione (**10**) > triflusal (**13**) > phenindione (**8**) > aspirin (**12**) > sodium citrate (**1**).

Most of the target drugs achieved feasible binding interactions with the target M^pro^. The poses experienced eligible binding scores and RMSD values were chosen for further investigation [32,33,34].

Idraparinux (**30**), fondaparinux (**29**), eptifibatide (**25**), and heparin (**23**) were found superior to the O6K inhibitor (**35**, docked). The aforementioned candidates (**23**, **25**, **29**, **30**, and **35**), besides ticagrelor (**26**), were selected for further investigation. Their 3D binding interactions within the M^pro^ binding pocket of SARS-CoV-2 are depicted in Table 1.

First, the binding of the docked O6K inhibitor within the dimeric M^pro^ of SARS-CoV-2 was analyzed. It was noted that O6K was stabilized by the formation of four H-bonds with residues Glu166, Asn142, Gly143, and Cys145 at 2.79, 3.04, 3.21, and 4.10 Å, respectively. The docking score was −9.35 kcal/mol and the root mean square deviation (RMSD) value was 1.41 Å. To evaluate the docking geometry correctness, the RMSD was used as a measure of the deviation degree of the docked co-crystallized ligand in relation to its original position (native co-crystallized ligand).

Idraparinux exhibited a score of −12.93 kcal/mol. It was also bound to the deep M^pro^ pocket through seven H-bonds with Pro168 (2 H-bonds), Asn214, Glu166, Asn142, Gln192, and Cys145 amino acids at 2.90, 3.48, 2.94, 2.98, 3.03, 3.47, and 3.86 Å, respectively.

The binding score of fondaparinux was −11.28 kcal/mol (RMSD = 2.26 Å), where one H-bond with Glu166 was calculated at 2.82 Å.

Eptifibatide showed a score of −10.31 kcal/mol at a value of RMSD = 2.01 Å. It exhibited six H-bonds with Gln192 (2 H-bonds), Glu166, Leu141, Met165, and Cys145 amino acids at 2.96, 3.25, 3.03, 3.24, 3.45, and 4.24 Å, respectively.

Heparin bound to the M^pro^ pocket of SARS-CoV-2 using seven H-bonds and displayed a docking binding score of −10.26 kcal/mol. It interacted with Thr25, Glu166, Asn142, His164, His163, Ser1, and Ser46 amino acids at 2.85, 2.97, 3.13, 3.25, 3.28, 3.39, and 3.47 Å, respectively.

Finally, ticagrelor experienced a score of -8.33 kcal/mol with only one H-bond formation with Gly143 at 3.54 Å.

### 2.2. Molecular Dynamics (MDs) Simulations Studies

To study the compounds’ stability with the best docking score at the M^pro^ active site of SARS-CoV-2, MD simulations were conducted for 200 ns. The obtained root mean square deviations (RMSDs) for the complexes and the ligands concerning their original positions within the active site were reported and analyzed. Frontier compounds’ interactions were also analyzed and evaluated in detail. Finally, the MM-GBSA free binding energy was estimated for all complexes during the simulation trajectories.

#### 2.2.1. Protein and Ligand RMSD Analysis

Five compounds were selected for MD simulations, including heparin, ticagrelor, fondaparinux, idraparinux, and eptifibatide, along with the co-crystallized inhibitor (O6K). The conformational stability of the proteins was monitored through the C_α_ atoms of the protein with respect to their initial position. As shown in Figure 3, all complexes showed stability during the 200 ns simulation, and the RMSD of all complexes was calculated at values less than 3.00 Å, which is acceptable for such proteins.

The RMSD was measured for ligands’ atoms based on the initial position of the heavy atoms inside the protein active site, as shown in Figure 4. The studied ligands showed early stability at the active site, except for ticagrelor, which left the active site ultimately at around 5.00 ns. Ticagrelor fluctuated till the end of the simulation. It was not able to stabilize within the active site and moved by 20.00 Å from its original site, as shown in Appendix A. On the other hand, other compounds showed much more stability during the simulation procedure. Fondaparinux was the most stable, but heparin along with the co-crystallized O6K inhibitor started fluctuating and moved around 8.00 Å from their original sites to become deeper inside the active site of the protease chain A of the dimer at about 70 ns, as shown in Appendix A. Idraparinux showed high stability inside the active site, barely moved by 4.00 Å at the beginning of the simulation, and fluctuated inside the active site. Finally, fondaparinux showed the most constancy inside the active site and moved by only 2.00 Å from its initial location.

Glu166 residue is a critical residue for the M^pro^ binding pocket of SARS-CoV-2; therefore, the distance between the ligands and this residue was measured throughout the simulation time using the measurement panel in Maestro software, as shown in Figure 5. Heparin moved by around 10 Å, while ticagrelor fluctuated and moved by approximately 25 Å and then stabilized at around a 15 Å distance. Idraparinux was unable to form hydrogen interactions with the Glu166 residue and kept a 4–5 Å distance during most of the simulation time. Eptifibatide showed stability in the first 25 ns; then, before it moved by around 10 Å till around 170 ns, it moved again to around 6 Å with regards to its original position. Fondaparinux and the co-crystallized inhibitor (O6K) showed similar behaviors. Both were around 6 Å from Glu166 before forming H-bonds at around 25 ns and 5 ns for fondaparinux and O6K, respectively. Both compounds held this H-bond till the end of the simulation time.

Since fondaparinux and idraparinux showed the most promising results, their interactions with the proteins will be discussed in detail.

#### 2.2.2. Histogram and Heat Map Analysis

Since fondaparinux and idraparinux showed the highest stability within the M^pro^ active pocket, their interactions are discussed in detail. 

Fondaparinux formed H-bond interactions with the following residues: Thr25 (~30%), Thr26 (~150%), His41 (~40%), Asn142 (~100%), Glu166 (~200%), Asp187 (~100%), and Gln189 (~250%), while idraparinux formed H-bonds with eight residues including Asn142 (~50%), Gly143 (~100), Ser144 (~30%), His163 (~70%), Glu166 (~200%), Thr169 (~50%), Thr190 (~50%), and Asn214 (~55%), as presented in Figure 6.

A percentage higher than 100 indicates formation of more than one H-bond interaction at the same time. Glu166 residue was able to form up to three H-bonds simultaneously with idraparinux, as shown in Figure 7.

Another type of H-bond interaction is the water-bridged H-bond, where crystal water molecules form a link between the protein residues and ligands. Fondaparinux formed water-bridged H-bonds with residues Thr24 (~40%), Ser46 (~30%), Asn142 (~50%), Gly143 (~150%), Glu166 (~50%), Arg188 (~50%), Gln189 (~50%), and Thr190 (~90%). While idraparinux was able to form more than 20 water-bridged H-bonds interactions, where bridge H-bonds with Thr26, Asn142, His164, Glu166, Pro168, Gln189, Gln192, Ser1, Asn214, and Asp216 held interactions more than 50% during the simulation time. Along with H-bonds interactions, fondaparinux was able to form a hydrophobic interaction with residue His41 (~90%) and an ionic interaction with Ser46 residue. On the other hand, idraparinux showed weak hydrophobic bonds with Leu27, Met165, and Met49, without the formation of ionic interactions.

Another method used to monitor these interactions involves plotting the number of interactions with respect to time. For example, a heat map (Figure 8) indicates the number of interactions at each frame, whereas the dark color indicates more interactions between proteins and ligands. Simulations revealed that fondaparinux holds interactions during the simulation time with Thr36, Asn142, Glu166, and Gln189, while idraparinux holds tight interactions with Gly143, Glu166, Ser1, and Asn214 residues.

### 2.3. MM-GBSA Calculations

The molecular mechanics, with a generalized Born and surface area solvation (MM–GBSA), were carried out to calculate both the ligand binding strain and free energies for docked ligands over the last 50 ns. The ΔG binding energies, Coulomb energies, hydrogen bond energies, generalized Born electrostatic energies, covalent binding energies, lipophilic energies, and van der Waals energies were recorded. The obtained results are described in Table 2 in more detail.

As shown in Table 2, fondaparinux and idraparinux showed high MM-GBSA binding energies with almost a 15 kcal/mol difference compared to the O6K inhibitor. In addition, idraparinux showed higher coulomb energy while fondaparinux showed higher H-bond energy.

In conclusion, MD suggests that both fondaparinux and idraparinux mechanisms of action mainly exist because of SARS-CoV-2 M^pro^ inhibition, while heparin and ticagrelor may have different mechanisms of action. Moreover, fondaparinux and idraparinux have good MM-GBSA binding energies, in agreement with the docking outputs, and support our claims that these compounds’ mode of action involves the M^pro^ inhibition of SARS-CoV-2.

### 2.4. Quantum Mechanics Studies

We utilized Gaussian software to calculate some molecular parameters of the most promising compounds in addition to the natural ligand of the receptor protein. Some selected molecular parameters are listed in Table 3. It was noted that idraparinux and fondaparinux have higher electronic energy, heat capacity, and entropy than other compounds, which attests to the electronic and thermal stability of the two drugs compared to other drugs. Polarizability, dipole moments, and HOMO-LUMO gaps are indicative of the change in the distribution of electrons over a molecule. It was also noted that idraparinux and fondaparinux have almost higher polarizability, dipole moment, and HOMO-LUMO gap values than other compounds.

It is well known that the outermost orbitals contribute most to the drug–receptor interactions; therefore, the charge density values of the highest occupied molecular orbital (HOMO) and the lowest unoccupied molecular orbital (LUMO) were calculated. The energy gap between HOMO and LUMO indicates the energy contribution to the covalent bond formation [35,36]. The graphical representations of the outermost orbitals’ electronic densities are listed in Table 4.

The molecular electrostatic potential (MEP) map helps to visualize the molecules’ charge distributions. The electron density mapping changes from high-density electrons (red color) to lower-density electrons (blue color). As described in Table 4, all promising compounds show areas of randomly distributed electrons, which may account for their ability to form multiple H-bonds which in turn enhance their binding energies with the atoms of the receptor.

### 2.5. In Vitro Studies

#### 2.5.1. The SARS-CoV-2 Inhibitory Assay (Cell-Based)

The attained CC_50_ for the selected screened anticoagulants (ticagrelor, fondaparinux sodium, dabigatran, and heparin) on Vero E6 cells (Figure 9) could pinpoint the safe concentrations for each investigated compound to be utilized in other in vitro assessments. The dose used to inhibit 50% of SARS-CoV-2 was also calculated for each screened compound by determining inhibitory concentration 50 (IC_50_), as shown in Figure 9. Notably, the attained IC_50_ values for the screened anticoagulant drugs were outstanding and consistent with their affinities and intrinsic activities calculated from the docking and molecular dynamics studies.

It is obvious that ticagrelor displays superior intrinsic activity to other drugs with an IC_50_ value of 5.6 µM and a safety index of 25.33. Additionally, **fondaparinux sodium** and **dabigatran** showed promising inhibitory activities as well, with IC_50_ values of 8.6 and 9.4 µM revealing safety indexes of 17.6 and 15.1, respectively. Moreover, **heparin** displayed inhibitory activity with an IC_50_ value of 105.9 µM and a safety index of 2.474. The CC_50_, IC_50_, and selectivity indexes of the screened compounds are listed in Table 5.

#### 2.5.2. SARS-CoV-2 M^pro^ Inhibitory Assay

The inhibitory potential of the outstanding anticoagulant candidates (ticagrelor, fondaparinux sodium, dabigatran, and heparin) towards the SARS-CoV-2 M^pro^ enzyme was investigated utilizing the SARS-CoV-2 M^pro^ assay. Moreover, tipranavir was used as a reference standard. Interestingly, promising SARS-CoV-2 M^pro^ inhibitory potential was attained for fondaparinux sodium with an IC_50_ value of 2.36 µM, surpassing the reference tipranavir (IC_50_ = 7.38 µM) by more than three-fold. Furthermore, highly eligible SARS-CoV-2 M^pro^ inhibitory potential was attained for dabigatran with an IC_50_ value of 10.59 µM. However, ticagrelor and heparin disclosed less weak SARS-CoV-2 M^pro^ inhibitory potentials with IC_50_ values of 47.53 and 152.7 µM, respectively, as depicted in Figure 10.

These results confirm the conclusion obtained from the MD simulations that the fondaparinux sodium mechanism of action is mainly because of SARS-CoV-2 M^pro^ inhibition, while heparin and ticagrelor may have different mechanisms of action. On the other hand, the M^pro^ inhibitory activity of dabigatran refers to the possibility of having more than one mechanism of action against SARS-CoV-2, besides the apparent M^pro^ inhibitory activity as well.

## 3. Structure–Activity Relationship (SAR) Study

Anticoagulant drugs can be classified according to their chemical structures and/or mechanism of action [37] into:
Anticoagulant:*Coumarins:* dicoumarol, phenprocoumon, acenocoumarol, warfarin, cyclocumarol, and ethyl biscoumacetate.*Indandiones:* phenandione, anisandione, bromindione, and diphenadione.*Injectable:* heparin, fondaparinux, and idraparinux.*Direct factor Xa inhibitors:* edoxaban, rivaroxaban, apixaban, and betrixaban. *Direct thrombin inhibitors (DTIs):* argatroban, ximelagatran, and dabigatran.Antiplatelets: aspirin, triflusal, dipyridamole, tirofiban, ticlopidine, clopidogrel, eptifibatide, ticagrelor, prasugrel, and cilostazole.Miscellaneous: sodium citrate, sulfinpyrazone, ethyl eicosapentaenoic acid, and defibrotide.

Based on the binding strength into the M^pro^ receptor of SARS-CoV-2 obtained from the molecular docking studies, we concluded the SAR of the tested anticoagulant drugs. Interestingly, the following results can be discussed (Figure 1 and Figure 11):
霂Collectively, among tested anticoagulant drugs, the injectable pentasaccharide anticoagulant heparin derivatives (compounds **29** and **30**), the sulfated mucopolysaccharides anticoagulant (compound **23**), as well as the antiplatelets with peptide-based structure (compound **25**) and nucleoside analog (compound **26**) showed the best binding strength against the M^pro^ of SARS-CoV-2, i.e., even better than the native co-crystallized O6K ligand (compound **35**).霂It is worth noting that the direct thrombin inhibitors (DTIs) retaining 4-carbamimidoyl moiety (compounds **20**, **31**, and **32**) showed better binding affinities than coumarins and indandione derivatives (compounds **8**, **9**, **10**, and **11**) against the M^pro^ receptor of SARS-CoV-2.霂Moreover, the studied SAR of coumarins revealed that the best activity was attained against the M^pro^ receptor of SARS-CoV-2 when a 4-hydroxy coumarin scaffold was attached to another 4-hydroxy coumarin scaffold through a methylene bridge (compound **7**).霂However, for indandiones, the best activity was attained when the scaffold was substituted with the bulky diphenylacetyl group at position 2 (compound **11**).霂Furthermore, for direct factor Xa inhibitors and anti-platelets, compounds **14** and **21** showed the best binding affinity in their class, respectively.

## 4. Materials and Methods

### 4.1. Molecular Docking Studies

Molecular docking studies were employed using MOE 2019.0102 software [38,39,40,41]. A set of 34 anticoagulant drugs was built and docked against the suggested active pocket of SARS-CoV-2 M^pro^. The main purpose of the studies was to evaluate the drug-protein interactions for the target compounds (1–34) in comparison to the co-crystallized inhibitor (O6K) (PDB ID: 6Y2G) [31].

#### 4.1.1. Validation of the Docking Results

To ensure that the used force field was suitable for the docking process, the co-crystal ligand (O6K) was redocked within the SARS-CoV-2 M^pro^ active site [42,43,44,45]. The obtained result showed that the docking procedure successfully regenerated the crystal pose with an RMSD value of 1.41 Å. The obtained poses are depicted in Figure 12, where the native ligand is shown in red and the redocked ligand is shown in green [46,47,48,49].

#### 4.1.2. The Target Anticoagulant Drugs Preparation

The PubChem database was employed for downloading the anticoagulant drug’s chemical structure. They were transferred from 2D to their corresponding 3D forms with respect to their stereochemistry configuration and examined for the atoms’ formal charges to be ready for docking [50,51,52,53]. Then, energy minimization was performed, followed by partial charge automatic calculation using the CHARMM27 force field [54,55,56,57]. Ultimately, they were imported into a single database along with the native co-crystallized O6K ligand and saved (MDB file) prior to the docking process within the target SARS-CoV-2 M^pro^ pocket, as described earlier [58,59,60,61].

#### 4.1.3. Target SARS-CoV-2 M^pro^ Preparation

The X-ray composition of the M^pro^ receptor of SARS-CoV-2 was attained by downloading from Protein Data Bank (PDB code: 6Y2G) [31]. Hence, the target protein became ready for the docking studies after considering the following steps: the backbones and side chains of the target protein were protonated with respect to their 3D geometry, the corrections were automatically utilized for the atom’s connection and type to check for any errors, and the receptor potential was fixed as discussed before [62,63,64,65]. CHARMM27 was implemented as the selected force field, and the native co-crystallized inhibitor active pocket was selected, as displayed before [66,67,68,69].

#### 4.1.4. Docking of the Anticoagulants to the SARS-CoV-2 M^pro^

Our set of 34 selected anticoagulant drugs and the native co-crystallized inhibitor (O6K) was employed for docking using MOE 2019.0102 software [70,71,72,73,74]. The MDB file was utilized for automatic docking. The docking methodology protocol was performed as before [75,76,77,78]. After the docking step ended, the attained poses were examined strictly, and the best pose was chosen for each ligand, as mentioned before in detail [79,80,81,82].

### 4.2. Molecular Dynamics (MDs) Studies

MD simulation studies were performed employing the Schrödinger LLC Desmond package [83]. Moreover, the energies of the molecular mechanics generalized Born surface area (MM-GBSA) were calculated for fondaparinux, idraparinux, and O6K complexes. The aforementioned MD methodologies are listed in detail in Appendix A [84,85,86].

### 4.3. Quantum Mechanics Studies

All structures were optimized using Gaussian 09, Revision C.01 software, and the B3LYP/3-21G method. The structures were further refined and optimized at B3LYP/6-31G and B3LYP/6-311G, and finally at B3LYP/6-311G*. No imaginary frequencies for the B3LYP/6-311G* optimized structures were obtained, confirming that the true local minima of the corresponding geometries were reached. The three-parameter hybrid exchange–correlation functional of Becke (B3LYP) [87,88], in combination with the 6-311G*, was also employed to run TD-DFT calculations. All quantum mechanics computations were implemented on Swinburne supercomputing facilities, utilizing GAUSSIAN 09 Revision C.0126 [89]. 

### 4.4. In Vitro Studies

#### 4.4.1. MTT Cytotoxicity Assay

Half of the maximal cytotoxic concentrations (CC_50_) were assessed to obtain the exact concentration of a compound that caused a rupture to 50% of cells after being treated with variant doses from the compounds to calculate the safety index (SI) = CC_50_/IC_50_ for each compound. The assay was performed through stock solution preparation of the target compounds in DMSO/ddH_2_O (for those in powder form). However, those in solution form were used directly. The prepared solutions were further diluted with DMEM by attaining the working solutions. Hence, the MTT methodology was employed to investigate the cytotoxic activity of the Vero E6 cell extracts, which will be used in other assays. Shortly, 96 well-plates were utilized for cell seeding and incubated at 37 °C in 5% CO_2_ for 24 h afterward. After one day, the incubated cells were subjected to different concentrations in triplicates of the investigated compounds. After another 24 h, the supernatant was dismissed, and sterile 1x phosphate buffer saline (PBS) was used for cell monolayer washing in triplicates. Thus, 20 µL of 5 mg/mL stock solution was added to each well and then incubated at 37 °C, followed by a medium aspiration for 4 h. So, the formed formazan crystals were dissolved by 200 µL isopropyl alcohol or DMSO. The formazan solution absorbance was measured at 540 nm (λ_max_) using a multi-well plate reader, while the reference wavelength was 620 nm. The CC_50_ of each target compound was measured employing nonlinear regression analysis of GraphPad Prism software [25].

#### 4.4.2. Crystal Violet Assay for the Determination of Inhibitory Concentration 50 (IC50)

The IC_50_ can be defined as the concentration needed to lower the viral-induced cytopathic effect (CPE) by 50% compared to the virus control. Hence, the IC_50_ for the investigated compounds was assessed, as formerly depicted [90], with slight modulations. Shortly, Vero E6 cells were placed and then incubated overnight in a 5% CO_2_ incubator at 37 °C. Thus, 1x PBS was used for cell monolayer washing. Thereafter, an aliquot of the virus (*h*CoV-19/Egypt/NRC-03/2020 (Accession Number on GSAID: EPI_ISL_430820)) following the tissue culture infection dose (TCID_50_) test was incubated with serially diluted compounds for 1 h. This step was used to determine the virucidal effects of the compounds before incubation with cells.

The cells were subjected to the virus–compound mix and then co-incubated at each well for 72 h in 100 µL. The untreated cells infected with the virus were considered “virus control”, but cells that were neither treated nor infected were considered “cell control”.

Cell fixation was carried out using 100 µL of 10% formaldehyde for 60 min and stained using 0.5 % crystal violet in distilled water at room temperature for 15 min. Thus, the crystal violet dye per well was dissolved using 180 μL of absolute methanol. The color’s optical density assessment was carried out from 570 to 620 nm with an Anthos Zenyth 200 rt plate reader (Anthos Labtec Instruments, Heerhugowaard, The Netherlands). The 50% inhibitory concentration (IC_50_) of each tested compound was evaluated by employing nonlinear regression analysis of GraphPad Prism software.

#### 4.4.3. SARS-CoV-2 M^pro^ Inhibitory Assay

The 3CL Protease Assay Kit was used to investigate the potential of the candidate anticoagulants (ticagrelor, fondaparinux sodium, dabigatran, and heparin) towards SARS-VoV-2 M^pro^ using DMSO as a solvent and employing the TECAN spark plate reader. The goal of the current assay was to evaluate the suggested mechanism of action for these outstanding anticoagulant members against SARS-CoV-2 M^pro^. The detailed applied protocol and methodology are depicted in the Appendix A.

## 5. Conclusions

Anticoagulant drugs showed very promising results for the discovery of effective drugs able to inhibit the M^pro^ of SARS-CoV-2 and reverse one of the major manifestations of COVID-19 (blood clotting). The 34 FDA-approved anticoagulant drugs were computationally studied utilizing molecular docking, molecular dynamics, and DFT approaches in addition to a biological evaluation of the most promising candidates. The molecular docking results revealed that fondaparinux and idraparinux are the most promising anticoagulants that can effectively bind to and inhibit the M^pro^ of SARS-CoV-2. In addition, MD simulations suggested that both fondaparinux and idraparinux mechanisms of action can be mainly implemented through SARS-CoV-2 M^pro^ inhibition, while heparin and ticagrelor may have different mechanisms of action. Moreover, fondaparinux and idraparinux have good MM-GBSA binding energies which agree with the docking outputs and support our claims that the mechanisms of action of these compounds involve the inhibition of the M^pro^ of SARS-CoV-2. Furthermore, quantum mechanics studies showed that both fondaparinux and idraparinux have almost higher polarizability, dipole moment, and HOMO-LUMO gap values than other compounds, which matches the higher binding energies of the two compounds over other studied compounds. Therefore, these findings open the door for proposing and understanding the atomic properties required to improve the binding affinity and strength of effective drug candidates toward SARS-CoV-2. Moreover, a biological evaluation of promising compounds was performed by carrying out the MTT cytotoxicity assay and the crystal violet assay to determinate the inhibitory concentration 50 (IC_50_). It was clear that ticagrelor displayed the best intrinsic activity with an IC_50_ value of 5.6 µM and a safety index of 25.33 against SARS-CoV-2. Fondaparinux sodium and dabigatran showed promising inhibitory activities as well, with IC_50_ values of 8.6 and 9.4 µM, revealing safety indexes of 17.6 and 15.1, respectively. Moreover, the SARS-CoV-2 M^pro^ inhibitory assay confirmed the potential activity of fondaparinux sodium, with an IC_50_ value of 2.36 µM surpassing the reference tipranavir (IC_50_ = 7.38 µM) by more than three-fold. Furthermore, a highly eligible SARS-CoV-2 M^pro^ inhibitory potential was attained for dabigatran with an IC_50_ value of 10.59 µM. This confirms the conclusion obtained from the MD simulations that the fondaparinux sodium mechanisms of action mainly exist because of SARS-CoV-2 M^pro^ inhibition, while heparin and ticagrelor may have different mechanisms of action. On the other hand, the M^pro^ inhibitory activity of dabigatran refers to the possibility of having more than one mechanism of action against SARS-CoV-2, besides the apparent M^pro^ inhibitory activity as well. Finally, the SAR study revealed that the injectable pentasaccharide anticoagulant derivatives (fondaparinux and idraparinux) showed the best binding strength against the M^pro^ receptor of SARS-CoV-2, even better than the native O6K inhibitor.

## Figures and Tables

**Figure 1 ijms-23-12235-f001:**
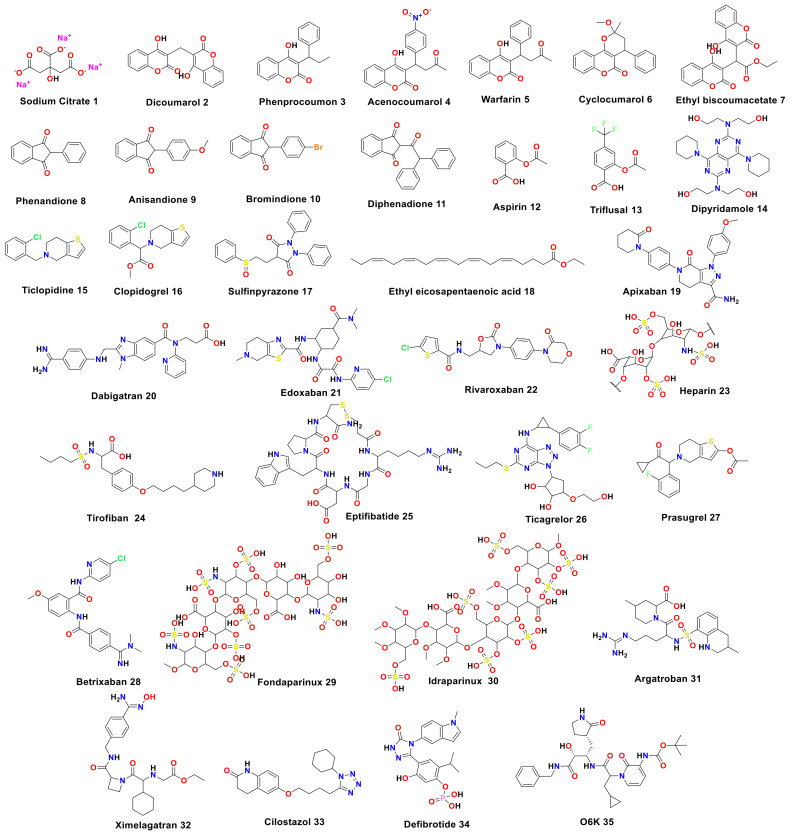
The chemical structures of selected anticoagulant drugs along with the co-crystallized inhibitor (O6K) 35 for M^pro^.

**Figure 2 ijms-23-12235-f002:**
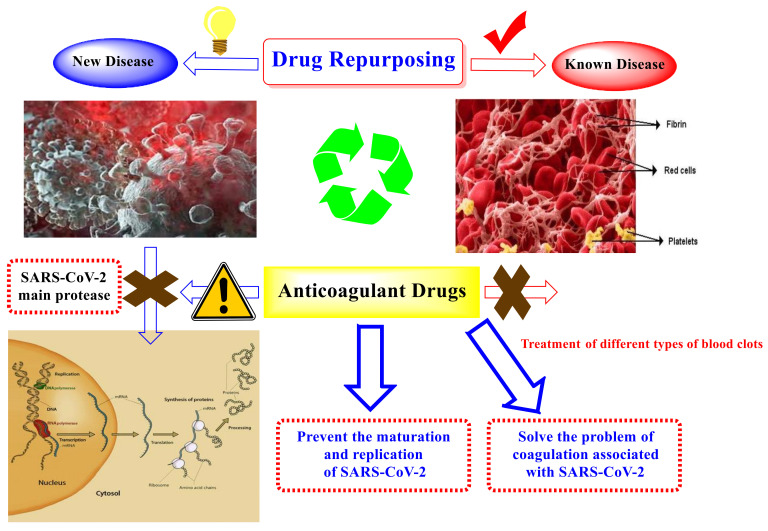
A graphical representation of the anticoagulants repositioned as SARS-CoV-2 inhibitors, showing their role in COVID-19 patient management.

**Figure 3 ijms-23-12235-f003:**
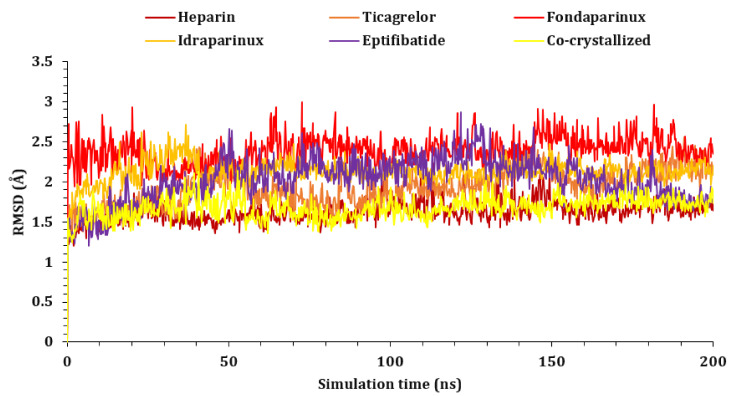
The RMSDs of protein complexes (heparin, ticagrelor, fondaparinux, idraparinux, eptifibatide, and O6K -6Y2G, respectively) for 200 ns.

**Figure 4 ijms-23-12235-f004:**
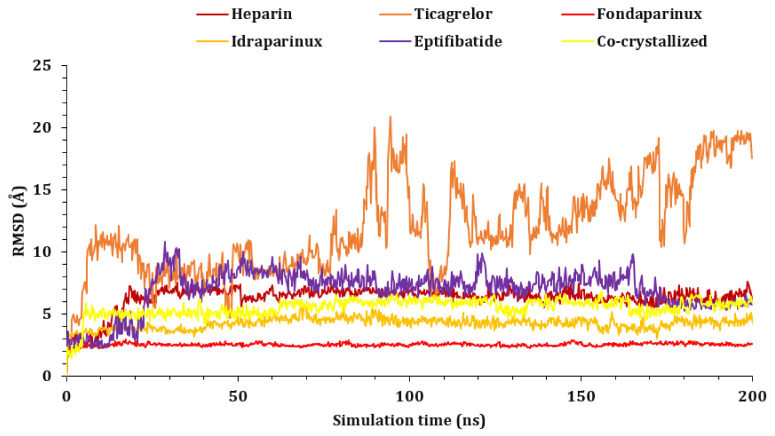
The RMSDs of ligand complexes (heparin, ticagrelor, fondaparinux, idraparinux, eptifibatide, and O6K-6Y2G, respectively) during the simulation time (200 ns).

**Figure 5 ijms-23-12235-f005:**
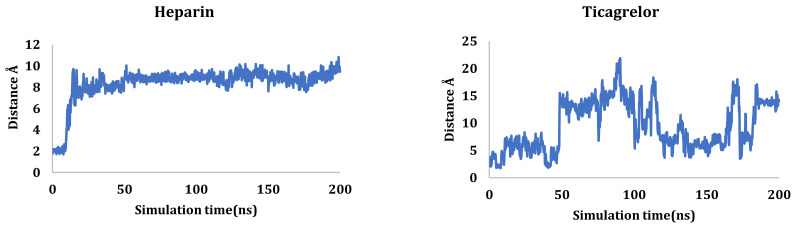
Distance in Å between Glu166 residue and ligands during the MD simulation using the measurement panel in Maestro software.

**Figure 6 ijms-23-12235-f006:**
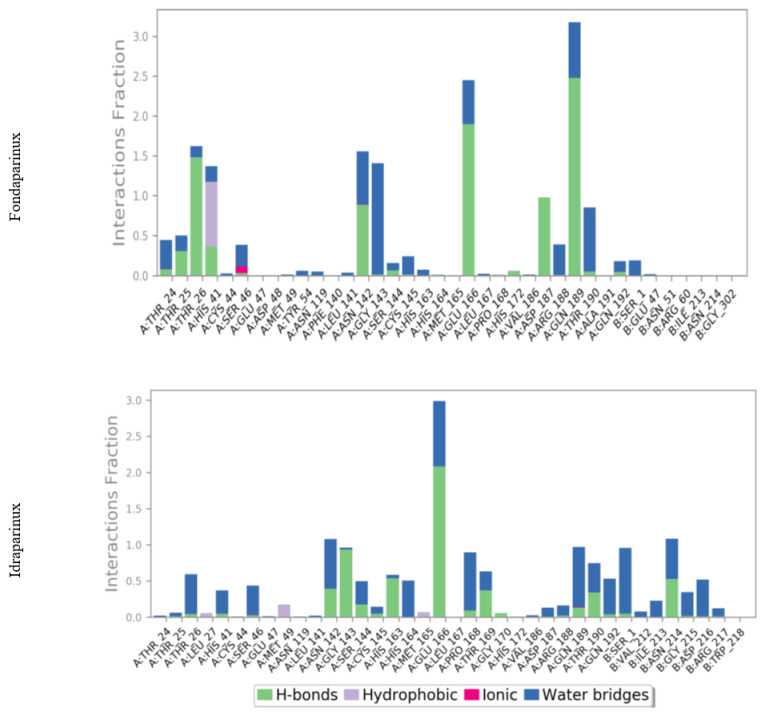
Histogram describing the binding interactions of fondaparinux and idraparinux within the SARS-CoV-2 M^pro^ during the simulation time (200 ns).

**Figure 7 ijms-23-12235-f007:**
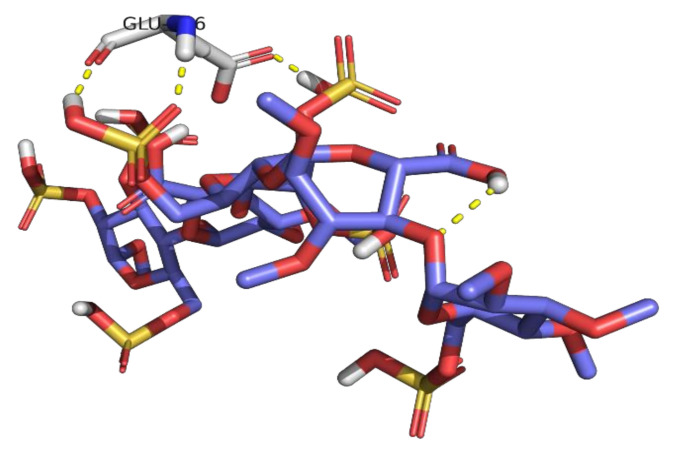
Hydrogen bond interactions between the protein residue (Glu166) and idraparinux.

**Figure 8 ijms-23-12235-f008:**
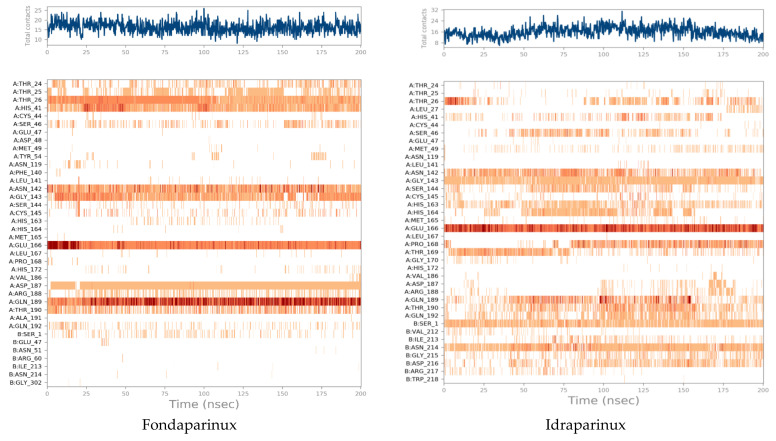
Heat map describing the total number of interactions within M^pro^ pocket during the 200 ns.

**Figure 9 ijms-23-12235-f009:**
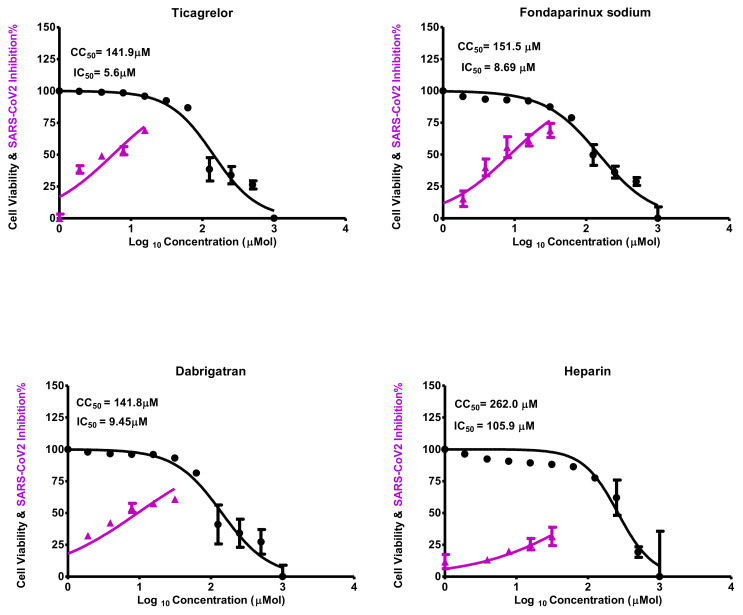
Cytotoxicity concentration 50 (CC_50_) and inhibitory concentration 50 (IC_50_) values for the screened anticoagulants (ticagrelor, fondaparinux sodium, dabigatran, and heparin), revealing their intrinsic activity against SARS-CoV-2. IC_50_ was calculated by the best line drawn between log concentration and viral inhibition % (triplicate for each concentration) by GraphPad Prism 5 software shown under the figure of CC_50_ and IC_50_.

**Figure 10 ijms-23-12235-f010:**
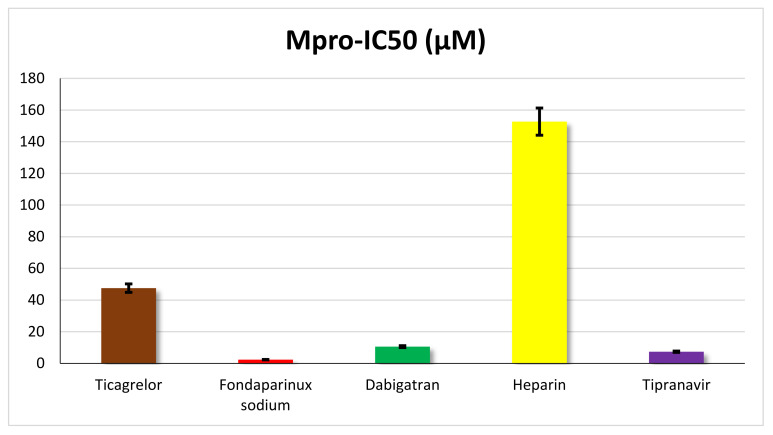
IC_50_ values for the outstanding anticoagulant candidates (ticagrelor, fondaparinux sodium, dabigatran, and heparin) towards the SARS-CoV-2 M^pro^ enzyme using tipranavir as a reference standard (triplicate for each concentration).

**Figure 11 ijms-23-12235-f011:**
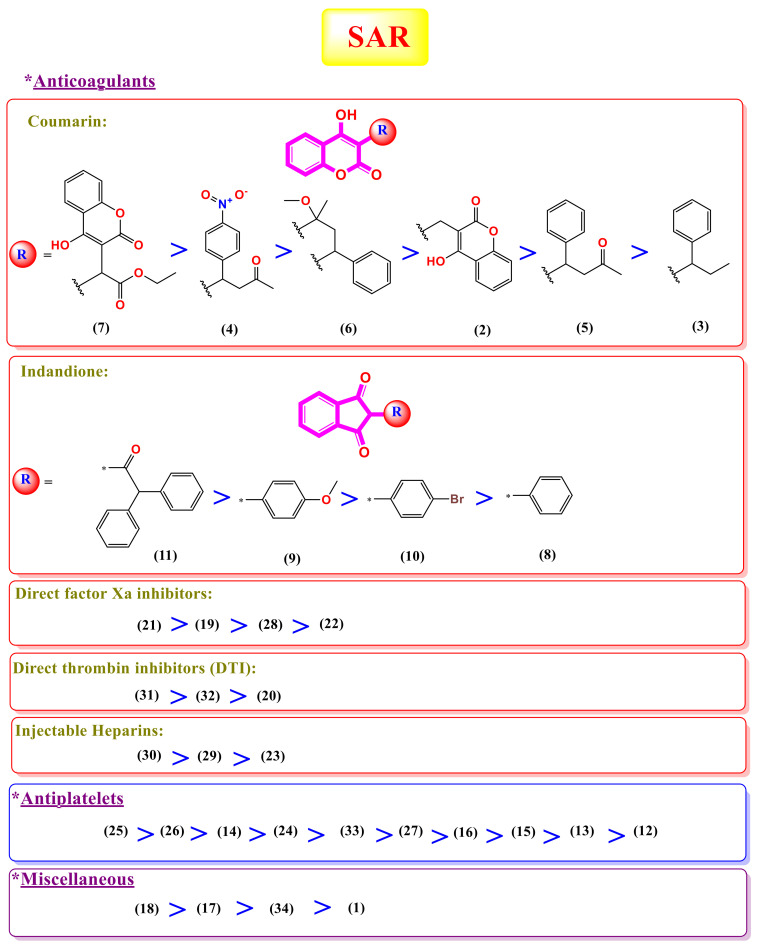
SAR study of the examined anticoagulant drugs towards SARS-CoV-2.

**Figure 12 ijms-23-12235-f012:**
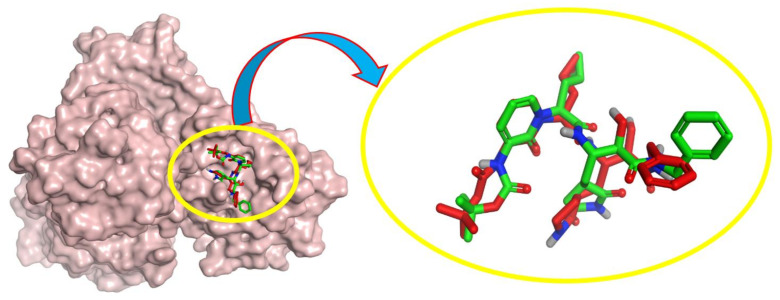
Superimposition of the redocked co-crystallized inhibitor O6K (green) over its native one (red).

**Table 1 ijms-23-12235-t001:** Three-dimensional binding interactions and positions of idraparinux, fondaparinux, eptifibatide, heparin, and ticagrelor compared to the O6K inhibitor within the binding site of SARS-CoV-2 M^pro^.

Rank	Drug	3D M^pro^ Interactions	3D M^pro^ Positions
1	Idraparinux	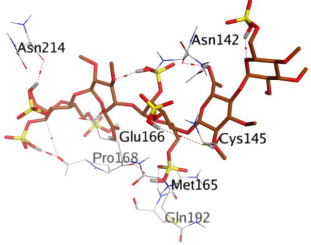	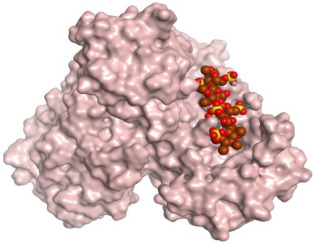
2	Fondaparinux	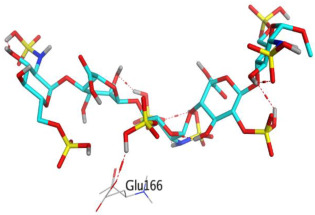	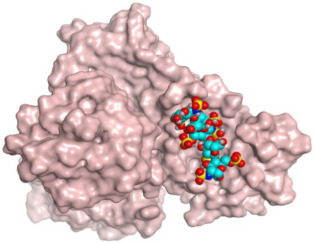
3	Eptifibatide	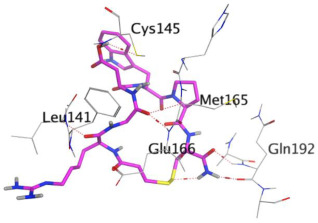	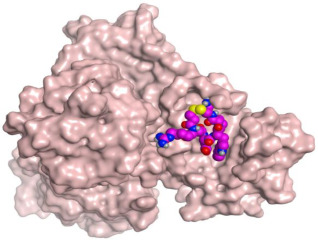
4	Heparin	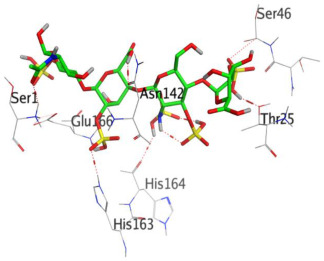	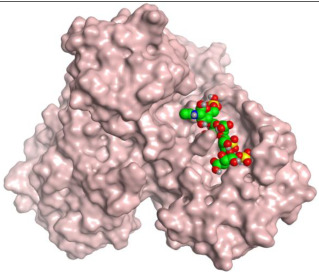
5	Ticagrelor	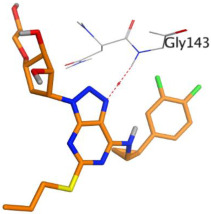	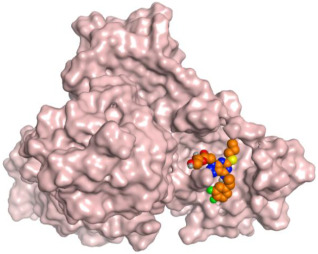
6	O6K	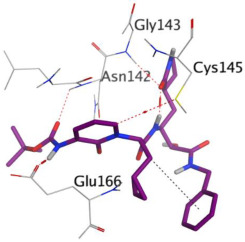	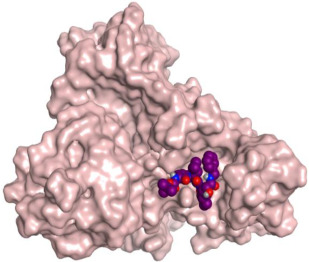

**Table 2 ijms-23-12235-t002:** MM-GBSA energies for (fondaparinux, idraparinux, and O6K-6Y2G) complexes of 6Y2G protein (kcal/mol).

Energies	Fondaparinux	Idraparinux	O6K
ΔG Binding	−51.24	−47.83	−66.14
Coulomb	−29.02	−33.05	−21.54
Covalent	1.51	3.00	1.25
H-bond	−5.59	−2.75	−1.90
Lipo	−4.09	−7.24	−15.07
Solv_GB	44.67	42.06	32.61
VdW	−58.72	−49.84	−59.75

Lipo: lipophilic energy; Solv_GB: generalized Born electrostatic solvation energy; vdW: van der Waals energy.

**Table 3 ijms-23-12235-t003:** Molecular parameters of the most promising compounds calculated at the B3LYP/6-311G* level of theory.

	Idraparinux	Fondaparinux	Eptifibatide	Heparin	Ticagrelor	O6K
Electronic energy (a.u)	−7960.687941	−8249.761481	−3439.760925	−5900.496951	−2120.921679	−2006.843019
EE + zero-point energy (a.u)	−7959.503380	−8248.720347	−3438.890071	−5899.677335	−2120.413977	−2006.140527
E (thermal) kcal/mol	804.301	710.103	582.073	558.193	340.739	467.855
Heat capacity (cal/mol-K)	357.951	342.236	214.894	262.334	130.586	162.591
Entropy(cal/mol-K)	516.009	476.337	321.245	387.736	232.036	268.442
Dipole moment (Debye)	7.6360	12.3118	8.7425	10.1676	1.4483	6.1357
Polarizability (a.u)	677.106290	639.424008	513.000082	492.779318	332.483453	388.563234
E_g_ (eV) *	6.62	6.31	4.26	5.73	4.80	4.55

* E_g_ refers to the energy gap between the HOMO and the LUMO.

**Table 4 ijms-23-12235-t004:** Charge densities of outermost orbitals and MEP map of the most promising compounds optimized at the B3LYP/6-311G* level of theory.

	HOMO	LUMO	MEP
Idraparinux	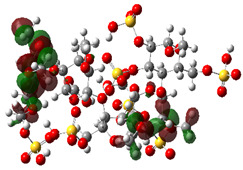	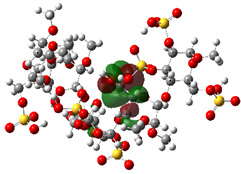	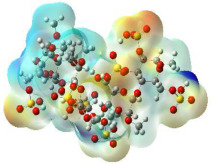
Fondaparinux	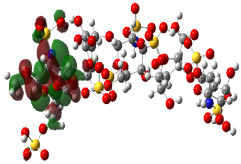	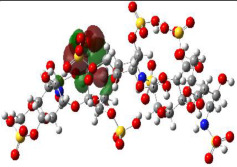	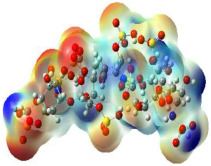
Eptifibatide	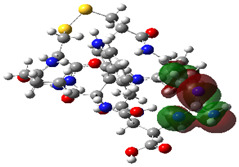	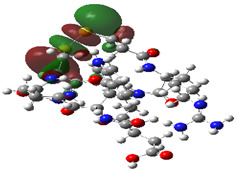	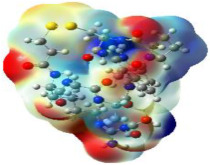
Heparin	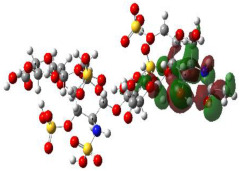	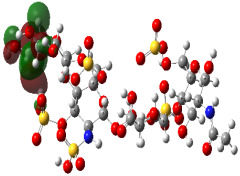	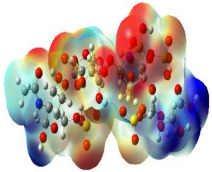
Ticagrelor	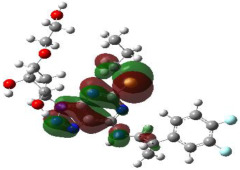	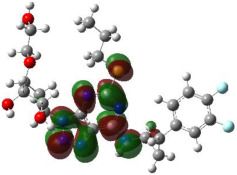	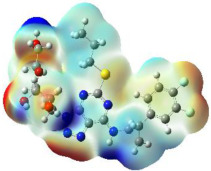
O6K	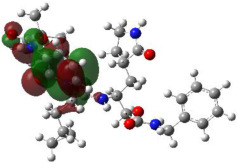	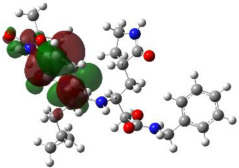	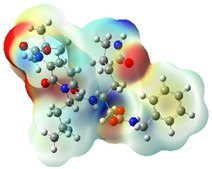

**Table 5 ijms-23-12235-t005:** Cytotoxicity and inhibitory effects of the screened anticoagulant drugs (ticagrelor, fondaparinux sodium, dabigatran, and heparin) against SARS-CoV-2 reveal their safety indexes.

Compound	Cytotoxicity Concentration (CC_50_) µM	Inhibitory Concentration (IC_50_) µM	Safety Index (SI)
Ticagrelor	141.90	5.60	25.33
Fondaparinux sodium	151.50	8.60	17.60
Dabigatran	141.80	9.40	15.10
Heparin	262.00	105.90	2.47

## Data Availability

Not applicable.

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
