# Peer review of "Anticoagulants as Potential SARS-CoV-2 Mpro Inhibitors for COVID-19 Patients: In Vitro, Molecular Docking, Molecular Dynamics, DFT, and SAR Studies"

_ijms, 2022, doi:10.3390/ijms232012235_

Round 1

Reviewer 1 Report

I would like to recommend a complete review of figure/plots legends. Some are superposed and a bit hard to view.

Author Response

Reviewer #1

Comments and Suggestions for Authors

I would like to recommend a complete review of figure/plots legends. Some are superposed and a bit hard to view.

Answer: The authors thank the reviewer for his effort to improve the quality of our manuscript. All figures/plots legends were reviewed and all figures were visualised to be clearer as requested. Regards.

Reviewer 2 Report

In the present study (ijms-1891072), the authors have made substantial in silico analysis to screen the binding affinity of 34 anticoagulants toward the main protease protein of SARs-CoV2 virus. The results revealed that, among investigated drugs, draparinux, fondaparinux, eptifibatide, heparin, and ticagrelor possess a high binding affinity. Subsequently, the authors have performed a molecular dynamic study to explore the dynamic and thermodynamic properties of the selected drugs.  Finally, the authors have conducted a cytotoxicity evaluation of these drugs which indicated that ticagrelor possesses the most potent cytotoxic activity with IC50 of 5.6uM.

Overall, this is an interesting study. However, the main aim of this study has NOT been achieved and/or proved. The main presented work is mainly based on computational analysis which must be completed/approved by at least in vitro binding activity toward Mpro enzyme. The authors have wrongly stated that these molecules have outstanding antagonist activity toward Mpro based on binding affinity, which is not right!! Further, the authors also stated that the cytotoxic activity of drugs indicates that this drug displays potent intrinsic potential for SARS-CoV-2, which is also a supported conclusion. Together, I would not suggest this study to be published in the presented form, unless extensive revision has been made. Here are some detailed concerns;

- the introduction is well-designed. The authors could support it with some recent reviews that detail molecular structure of SARs-Cov2 e.g., doi.org/10.3390/pharmaceutics13111759

- As mentioned before, the authors must perform and present in vitro exploration of the selected best binding drugs toward the activity of Mpro enzyme. Otherwise, this study is not solid enough and the presented conclusion would be not supported. The mode of action is really requested for IJMS scope!! Otherwise, the authors can transfer the manuscript to other MDPI journals that mainly focus on in silico investigations, as Molecules.

- It is astounding to see that the authors consider the high binding affinity as high antagonist activity!!This totally wrong!! antagonist activity is mainly based on in vitro/in vivo assessments toward the targeted molecules. The binding affinity of drug does NOT indicate that this drug possesses an antagonist. Accordingly, the authors must avoid stating that the high binding affinity reveal that these drugs have the most outstanding antagonistic activity on Mpro.

- Although, the authors have stated that the molecular docking is matching with the cytotoxic activity, ticagrelor showed a really low binding affinity (-8.33 kcal/mol), as compared to the other drugs, which is mainly based on only ONe H-binding to Gly143. However, ticagrelor showed the best cytotoxic activity!! this is contrary to the authors' statement!! line 375

- Although, the authors stated that in silico screening revealed that idraparinux possessed a promising binding affinity, the authors have excluded this compound for in vitro cytotoxicity and replace it with Dabrigatran!!!?? this is totally strange!!

- correct IC50/CC50 diagrams to uM not umol and add SD for all figures.

- Although, Eptifibatide indicated a high binding affinity and good interactions to Mpro as O6K ligand, it was not clear based on what the authors have decided to exclude this drug from molecular dynamic studies and in vitro cytotoxicity??

- Regarding the presented SAR study is really poorly understandable. The authors should discuss it in better way. Further, it is suggested that the authors move this part together with molecular modelling data.

- finally, the manuscript requires moderate language editing for proper academic style and typos mistakes. e.g. in abstract (Quantum mechanical studies were also conducted which helped to at test to some of the molecular docking and dynamics findings.

Author Response

Reviewer #2

Comments and Suggestions for Authors

In the present study (ijms-1891072), the authors have made substantial in silico analysis to screen the binding affinity of 34 anticoagulants toward the main protease protein of SARs-CoV2 virus. The results revealed that, among investigated drugs, draparinux, fondaparinux, eptifibatide, heparin, and ticagrelor possess a high binding affinity. Subsequently, the authors have performed a molecular dynamic study to explore the dynamic and thermodynamic properties of the selected drugs.  Finally, the authors have conducted a cytotoxicity evaluation of these drugs which indicated that ticagrelor possesses the most potent cytotoxic activity with IC50 of 5.6uM.

Overall, this is an interesting study. However, the main aim of this study has NOT been achieved and/or proved. The main presented work is mainly based on computational analysis which must be completed/approved by at least in vitro binding activity toward Mpro enzyme. The authors have wrongly stated that these molecules have outstanding antagonist activity toward Mpro based on binding affinity, which is not right!! Further, the authors also stated that the cytotoxic activity of drugs indicates that this drug displays potent intrinsic potential for SARS-CoV-2, which is also a supported conclusion. Together, I would not suggest this study to be published in the presented form, unless extensive revision has been made. Here are some detailed concerns;

Answer: The authors thank the reviewer for his efforts to improve the quality of our manuscript. Thanks again for your positive impression of our study. All the requested modifications were considered with great interest to gain the satisfaction of the respected reviewer. Briefly, the in vitro binding activity of the examined candidates toward Mpro enzyme of SARS-CoV-2 was carried out according to the reviewer's request.

- the introduction is well-designed. The authors could support it with some recent reviews that detail molecular structure of SARs-Cov2 e.g., doi.org/10.3390/pharmaceutics13111759

Answer: Thanks a lot for your supportive comment. More details about SARS-CoV-2 molecular structure were discussed in the introduction as requested citing some references such as doi.org/10.3390/pharmaceutics13111759.

- As mentioned before, the authors must perform and present in vitro exploration of the selected best binding drugs toward the activity of Mpro enzyme. Otherwise, this study is not solid enough and the presented conclusion would be not supported. The mode of action is really requested for IJMS scope!! Otherwise, the authors can transfer the manuscript to other MDPI journals that mainly focus on in silico investigations, as Molecules.

Answer: The authors agree with the reviewer for his recommendation. Therefore, the in vitro binding activity of the examined candidates toward Mpro enzyme was carried out and discussed in the manuscript according to the reviewer's request.   

- It is astounding to see that the authors consider the high binding affinity as high antagonist activity!!This totally wrong!! antagonist activity is mainly based on in vitro/in vivo assessments toward the targeted molecules. The binding affinity of drug does NOT indicate that this drug possesses an antagonist. Accordingly, the authors must avoid stating that the high binding affinity reveal that these drugs have the most outstanding antagonistic activity on Mpro.

Answer: The authors thank the reviewer for his comment. Ok, all the stated (antagonistic activity) was removed from the in silico parts and replaced only by (binding affinity) as requested.

- Although, the authors have stated that the molecular docking is matching with the cytotoxic activity, ticagrelor showed a really low binding affinity (-8.33 kcal/mol), as compared to the other drugs, which is mainly based on only ONe H-binding to Gly143. However, ticagrelor showed the best cytotoxic activity!! this is contrary to the authors' statement!! line 375

Answer: The authors thank the reviewer for his observation. However, the authors carried out the molecular docking on the SARS-CoV-2 Mpro target which revealed that ticagrelor showed a low binding affinity (-8.33 kcal/mol) than idraparinux, fondaparinux, eptifibatide, and heparin (But still very close to that of the co-crystallized O6K inhibitor, -9.35 kcal/mol). On the other hand, the in vitro studies were performed on the Vero E6 cells of SARS-CoV-2 (NOT Mpro target) which showed that ticagrelor achieved the best cytotoxic activity. Also, the molecular dynamics simulations confirmed the lower affinity of ticagrelor towards the SARS-CoV-2 Mpro. Therefore, there is no conflict between the two discussed results, and the superior anti-SARS-CoV-2 activity of ticagrelor seems to be due to a different mechanism of action targeting another target other than Mpro. Finally, we tried to explain this inside the manuscript by saying that (In conclusion, MD suggests that both fondaparinux and idraparinux mechanism of action is through the SARS-CoV-2 Mpro inhibition, while heparin and ticagrelor may have a different mechanism of action).Regards.

- Although, the authors stated that in silico screening revealed that idraparinux possessed a promising binding affinity, the authors have excluded this compound for in vitro cytotoxicity and replace it with Dabrigatran!!!?? this is totally strange!!

Answer: The authors thank the reviewer for his deep observation. First, the authors did not exclude idraparinux from the in vitro cytotoxicity but we could not get its standard to be tested biologically among the selected compounds. Therefore, we completed its deep in silico studies using the MD simulations to confirm its better docking result. Second, we added dabigatran to the biologically tested candidates although not being one of the superior members against the SARS-CoV-2 Mpro target was intentional. This was done to confirm the principle concept of the study that most anticoagulants may have anti-SARS-CoV-2 activity although not superior to SARS-CoV-2 Mpro inhibition. Regards.         

- Correct IC50/CC50 diagrams to uM not umol and add SD for all figures.

Answer: Ok, IC50/CC50 diagrams were adjusted o uM not umol as requested and the figures’ legends were modified to indicate the SD.

- Although, Eptifibatide indicated a high binding affinity and good interactions to Mpro as O6K ligand, it was not clear based on what the authors have decided to exclude this drug from molecular dynamic studies and in vitro cytotoxicity??

Answer: The authors thank the reviewer for his comment. Now, eptifibatide was included in the molecular dynamics studies as requested.

- Regarding the presented SAR study is really poorly understandable. The authors should discuss it in better way. Further, it is suggested that the authors move this part together with molecular modelling data.

Answer: The SAR part was discussed in a better way as requested. We hope it becomes more clear and better understandable. Thanks in advance.

- Finally, the manuscript requires moderate language editing for proper academic style and typos mistakes. e.g. in abstract (Quantum mechanical studies were also conducted which helped to at test to some of the molecular docking and dynamics findings.

Answer: Ok, all typos were corrected as requested. Moreover, the whole manuscript’s language was revised thoroughly by a native English speaker. Thanks in advance.

Reviewer 3 Report

In this manuscript, 34 anticoagulants were in silico screened against the main protease Mpro of SARS-CoV-2 inhibitors, including molecular dockings, MD simulations, and quantum mechanics studies. The biological evaluation was also performed to assess their inhibitory activity. It is a comprehensive study with drug repurposing, however, it needs a lot of improvements before publication.

 Major issues:

1.     The introduction is short and lacks important information and logicality. First, the description of Mpro, the target, is too brief. And it is weird that the topic turns to drug repurposing and turns back to Mpro. More about the target is expected to be introduced. The last paragraph should be improved to clearly express the specific purpose of this work instead of just a big goal. Ideally, readers could quickly learn about what you have done and the expectations of your paper. But now the message conveyed in this paragraph is very vague.

2.     The binding interaction figures in Table 1 need to be regenerated since the labeled residues are too small and some are obscured by the sticks. The interactions could not be observed clearly in the present figures.

3.     In the molecular dockings, authors gave an order of the binding strengths of these anticoagulants based on their docking scores: idraparinux (30) > fondaparinux (29) > eptifibatide (25) > heparin (23) > ticagrelor (26) > O6K inhibitor (35). I noticed that O6K formed four hydrogen bonds with the residues and had a docking score of -9.35 kcal/mol. But the docking score of ticagrelor is -8.33 kcal/mol (Line 238). I wonder if there is a typo or a mistake.

Both fondaparinux and ticagrelor formed only one hydrogen bond with a surrounding residue, but their binding strengths are considered as stronger than O6K in the above order. Could you explain this to readers?

4.     How the distance between a specific residue and the ligands were calculated? How the H-bond interaction fractions with the residues were calculated? Some details are suggested to be implemented in the method section.

5.     The authors mentioned that ticagrelor displayed the strongest activity with an IC50 of 5.6 µM among the tested four compounds: ticagrelor, fondaparinux sodium, dabigatran, and heparin. However, ticagrelor showed unsatisfactory MD simulation results. It was the least stable in the active site. Were the simulation results really in compliance with their actual activities? Some explanations may help to address the point of inconsistency. The simulation results reveal that idraparinux is also a highly promising inhibitor. Did you perform the biological evaluation of it?

Minor points:

1.     Line 103, which is the suggested active pocket?

2.     Line 111, the unit of RMSD is missing.

3.     Line 127, "It got ready…" is not a formal expression.

4.     Line 138, what do you mean by "the best conformers for each ligand were chosen as mentioned before in detail"? The sentence needs to be revised.

5.     Line 211, what do you mean by "almost comparable binding poses"? What are the similarities and differences?

6.     Line 216, the number "36" is inconsistent with the aforementioned candidate O6K inhibitor (35).

7.     Line 224, the RMSD here needs to be explained.

8.     In Figure 5, the illustrations of all the five systems should be given as in Figure 4.

9.     In this journal, the results section should be presented in front of the methods section.

There are many grammatical errors and oral expressions in the manuscript and the authors are advised to do some language-editing work to make the text more readable.

Author Response

Reviewer #3

Comments and Suggestions for Authors

In this manuscript, 34 anticoagulants were in silico screened against the main protease Mpro of SARS-CoV-2 inhibitors, including molecular dockings, MD simulations, and quantum mechanics studies. The biological evaluation was also performed to assess their inhibitory activity. It is a comprehensive study with drug repurposing, however, it needs a lot of improvements before publication.

Answer: The authors thank the reviewer for his efforts to improve the quality of our manuscript. All the requested modifications were considered with great intention to gain the satisfaction of the respected reviewer.

 Major issues:

  1. The introduction is short and lacks important information and logicality. First, the description of Mpro, the target, is too brief. And it is weird that the topic turns to drug repurposing and turns back to Mpro. More about the target is expected to be introduced. The last paragraph should be improved to clearly express the specific purpose of this work instead of just a big goal. Ideally, readers could quickly learn about what you have done and the expectations of your paper. But now the message conveyed in this paragraph is very vague.

Answer: A detailed description of Mpro was added to the introduction part as requested. Moreover, the last paragraph of the introduction was rephrased to be much clear according to the reviewer’s comment.

  1. The binding interaction figures in Table 1 need to be regenerated since the labeled residues are too small and some are obscured by the sticks. The interactions could not be observed clearly in the present figures.

Answer: The authors thank the reviewer for his observation, all Figures in Table 1 were regenerated to be clearer as requested. Now, the interactions could be observed clearly in the present Figures. Regards.  

  1. In the molecular dockings, authors gave an order of the binding strengths of these anticoagulants based on their docking scores: idraparinux (30) > fondaparinux (29) > eptifibatide (25) > heparin (23) > ticagrelor (26) > O6K inhibitor (35). I noticed that O6K formed four hydrogen bonds with the residues and had a docking score of -9.35 kcal/mol. But the docking score of ticagrelor is -8.33 kcal/mol (Line 238). I wonder if there is a typo or a mistake.

Answer: The authors thank the reviewer for his deep observation. It was a typo error as the docking score for O6K (-9.35 kcal/mol) is larger than that of ticagrelor (-8.33 kcal/mol). Now, the order of the binding strengths was corrected as requested.

Both fondaparinux and ticagrelor formed only one hydrogen bond with a surrounding residue, but their binding strengths are considered as stronger than O6K in the above order. Could you explain this to readers?

Answer: The authors thank the reviewer for his comment. However, the docking score refers to the binding affinity of a specific residue towards the binding pocket. Therefore, as the binding score increases, the stability of the compound inside the binding pocket increases as well. This does not depend on the number of H-bonds formed as a compound with a higher binding score and large affinity towards the receptor pocket may require a lower number of interactions to be able to fix itself within the receptor pocket. Briefly, the docking score is the main issue to be able to rank the examined compounds according to their binding strengths. Regards.  

  1. How the distance between a specific residue and the ligands were calculated? How the H-bond interaction fractions with the residues were calculated? Some details are suggested to be implemented in the method section.

Answer: The authors thank the reviewer for his comment. The distance between a specific residue and the ligands was calculated using the measurement panel in Maestro software (added to the manuscript). On the other hand, the interaction histogram was plotted using Maestro software's simulation interaction diagram (SID) panel. This histogram presents the interaction as a percentage: a compound that forms a single interaction with a residue throughout the stimulation time will have 100% interaction. Therefore, the stacked bar charts are normalized throughout the trajectory: for example, a value of 0.7 suggests that 70% of the simulation time the specific interaction is maintained. Moreover, values over 1.0 are possible as some protein residue may make multiple contacts of the same subtype with the ligand.

  1. The authors mentioned that ticagrelor displayed the strongest activity with an IC50of 5.6 µM among the tested four compounds: ticagrelor, fondaparinux sodium, dabigatran, and heparin. However, ticagrelor showed unsatisfactory MD simulation results. It was the least stable in the active site. Were the simulation results really in compliance with their actual activities? Some explanations may help to address the point of inconsistency. The simulation results reveal that idraparinux is also a highly promising inhibitor. Did you perform the biological evaluation of it?

 Answer: The authors thank the reviewer for his observations. However, the authors carried out the molecular docking on the SARS-CoV-2 Mpro target which revealed that ticagrelor showed a low binding affinity (-8.33 kcal/mol) than idraparinux, fondaparinux, eptifibatide, and heparin (But still very close to that of the co-crystallized O6K inhibitor, -9.35 kcal/mol). Accordingly, its MD simulation is not promising due to its low docking binding affinity (-8.33 kcal/mol) than others. On the other hand, the in vitro studies were performed on the Vero E6 cells of SARS-CoV-2 (NOT Mpro target) which showed that ticagrelor achieved the best cytotoxic activity. Also, the molecular dynamics simulations confirmed the lower affinity of ticagrelor towards the SARS-CoV-2 Mpro. Therefore, there is no conflict between the two discussed results, and the superior anti-SARS-CoV-2 activity of ticagrelor seems to be due to a different mechanism of action targeting another target other than Mpro. Finally, we tried to explain this inside the manuscript by saying that (In conclusion, MD suggests that both fondaparinux and idraparinux mechanism of action is through the SARS-CoV-2 Mpro inhibition, while heparin and ticagrelor may have a different mechanism of action).

On the other hand, the authors did not exclude idraparinux from the in vitro cytotoxicity but we could not get its standard to be tested biologically among the selected compounds. Therefore, we completed its deep in silico studies using the MD simulations to confirm its better docking result.  

Minor points:

  1. Line 103, which is the suggested active pocket?

Answer: The whole paragraph was rephrased to be much clear.

  1. Line 111, the unit of RMSD is missing.

Answer: The unit of RMSD was Angstrom (Å) and that was added as requested.

  1. Line 127, "It got ready…" is not a formal expression. 

Answer: The sentence "It got ready…" was rephrased to be more formal.

  1. Line 138, what do you mean by "the best conformers for each ligand were chosen as mentioned before in detail"? The sentence needs to be revised.

Answer: The sentence was revised and rephrased to be much clear.

  1. Line 211, what do you mean by "almost comparable binding poses"? What are the similarities and differences?

Answer: The sentence was rephrased to eliminate ambiguity.

  1. Line 216, the number "36" is inconsistent with the aforementioned candidate O6K inhibitor (35).

Answer: The number 36 was replaced by the number 35 as requested.

  1. Line 224, the RMSD here needs to be explained. 

Answer: RMSD was explained as requested.

  1. In Figure 5, the illustrations of all the five systems should be given as in Figure 4.

Answer: Ok, Figure 5 was corrected as requested.

  1. In this journal, the results section should be presented in front of the methods section.

 Answer: Ok, the results section was moved to be presented in front of the methods section as requested. Thanks.  

There are many grammatical errors and oral expressions in the manuscript and the authors are advised to do some language-editing work to make the text more readable.

Answer: Ok, all typos were corrected and the whole manuscript’s language was revised thoroughly by a native English speaker.

Round 2

Reviewer 2 Report

the authors have adequately addressed major concerns that have been raised in the first report and the manuscript has been significantly improved. However, there are few minor concerns that the authors should revise:

- regarding the newly added Mpro inhibition, the authors should present the results in triplicated and add SD/SE values (Fig10).

- this also should apply in Fig9, it seems that several spots in fig without SD.

- the SARS-CoV-2 Mpro inhibitory assay should be detailed in the main manuscript, not in SI.

Author Response

                                                      Reviewer #2

Comments and Suggestions for Authors

The authors have adequately addressed major concerns that have been raised in the first report and the manuscript has been significantly improved. However, there are few minor concerns that the authors should revise:

Answer: The authors thank the reviewer for his efforts to improve the quality of our manuscript. Thanks again for your positive impression of our study. All the requested modifications were considered according to the reviewer’s comments.

- regarding the newly added Mpro inhibition, the authors should present the results in triplicated and add SD/SE values (Fig10).

Answer: The authors thank the reviewer for his comment. The results of the newly added Mpro inhibitory assay were obtained already from triplicated measurements. The SD/SE values were added to Figure 10 as requested.

- this also should apply in Fig9, it seems that several spots in fig without SD.

Answer: The authors thank the reviewer for his comment. All points in Figure 9 were adjusted by SD as requested.

- the SARS-CoV-2 Mpro inhibitory assay should be detailed in the main manuscript, not in SI.

Answer: Ok, more details about the SARS-CoV-2 Mpro inhibitory assay were added as requested. However, the detailed protocol and methodology are very long and reported previously in the literature, so the authors kept them in the SI to be helpful for others. Thanks in advance.